# Heterologous Expression of a Ferritin Homologue Gene *PpFer1* from *Prunus persica* Enhances Plant Tolerance to Iron Toxicity and H_2_O_2_ Stress in *Arabidopsis thaliana*

**DOI:** 10.3390/plants12244093

**Published:** 2023-12-07

**Authors:** Yong Yang, Jinjin Zhang, Mengyuan Li, Youzheng Ning, Yifei Tao, Shengpeng Shi, Adeeba Dark, Zhizhong Song

**Affiliations:** 1Zhenjiang Academy of Agricultural Sciences, Zhenjiang Institute of Agricultural Sciences in Hilly Areas of Jiangsu Province, Zhenjiang 212400, China; yl0656@163.com; 2The Engineering Research Institute of Agriculture and Forestry, Ludong University, No. 186 Hongqizhong Road, Yantai 264025, China; LMY04826@outlook.com (M.L.); 19856293187@163.com (Y.T.); 3Faculty of Modern Agriculture, Linyi Vocational University of Science and Technology, No. 1 Macau Road, Linyi 276000, China; lingyunmaomao@163.com; 4Department of Plant Science, University of Cambridge, Cambridge CB2 3EA, UK; yn283@cam.ac.uk (Y.N.); ss2619@cam.ac.uk (S.S.); amd57@cam.ac.uk (A.D.); 5Wolfson College, University of Cambridge, Cambridge CB3 9BB, UK

**Keywords:** peach, Fe storage, ferritin, Fe toxicity, H_2_O_2_ stress

## Abstract

In plants, ferritin proteins play an important role in iron (Fe) storage which contributes to plant growth and development. However, the biological functions of ferritins in fruit trees are essentially unknown. In this study, three *Ferritin* genes were isolated from ‘Zhentong No. 3’ peach, which were named *PpFer1-PpFer3*. The expression levels of these genes were different in distinct tissues/organs. Notably, *PpFer1* was the most abundantly expressed Ferritin family gene in all tested tissues of ‘Zhentong No. 3’ peach; its expression levels were significantly enhanced throughout the entire peach seedling under Fe toxicity and H_2_O_2_ stress, particularly in the leaves. In addition, over-expression of *PpFer1* was effective in rescuing the retarded growth of *Arabidopsis fer1-2* knockout mutant, embodied in enhanced fresh weight, primary root length, lateral root numbers, total root length, total leaf chlorophyll, stomatal conductance (*G*_s_), net photosynthetic rate (*P*_n_), transpiration rate, and tissue Fe concentration. This study provides insights into understanding the molecular mechanisms of Fe storage and sequestration in perennial fruit trees.

## 1. Introduction

Iron (Fe) is one of the abundant mineral elements in plant cells, and it participates in many metabolic pathways and life processes, such as photosynthesis, respiration, hormone synthesis, energy metabolism, and DNA repair [1,2,3,4]. In particular, Fe deficiency in soils causes serious crop yield decrease and quality reduction, whereas excessive Fe may impair plant growth and cause soil pollution [5,6,7]. Therefore, plants need to take advantage of accurate Fe uptake, transport, storage, and utilization strategies to maintain normal growth and development.

In higher plants, two kinds of root Fe absorption strategies have been identified, especially under Fe deficiency stress [7,8,9,10,11]. Strategy I is found in dicotyledons and non-gramineous monocotyledons, in which Fe^3+^ is reduced to Fe^2+^ through ferric reduction oxide (FRO), and Fe^2+^ is absorbed by iron regulated transporters (IRT). Strategy II is observed in gramineous plants, in which Fe^3+^ is absorbed through a specific type of Fe^3+^ chelator phytosideophore (PS) pathway that depends on yellow-stripe (YS) or yellow-stripe-like (YSL) transporters [10,11,12]. 

The mobilization of intracellular Fe^2+^ is crucial for plant growth and development, especially under Fe-deficient conditions. When Fe^2+^ enters the cell, it needs to be transported to each organelle for distribution and utilization or stored to form an intracellular Fe^2+^ pool. However, when Fe^2+^ in the cytoplasm is excessively stored, it will also cause Fe^2+^ toxicity which affects plant growth. Previous studies showed that transporter proteins, like natural-resistance-associated macrophage proteins (NRAMPs), permease in chloroplast (PIC), and vacuolar iron transporters (VIT), are involved in the transport and distribution of Fe^2+^ within plant cells [7,8,9,10,11]. The plastid can act as an Fe^2+^ pool in the cell, sensing and regulating the concentration of Fe^2+^ to adapt to changes in external Fe supply [7,11,13]. In seeds of *Arabidopsis thaliana*, the globoids in the vacuoles are the main storage pool for Fe^2+^, containing approximately 50% of total Fe^2+^. Typically, intracellular Fe^2+^ is favorably stored in vacuoles and may also be chelated into ferritin, which is utilized in various Fe^2+^-dependent metabolic pathways or physiological processes. In particular, ferritin is a type of 24-protein polymer encoded by nuclear genes, which has a highly conserved structure in eukaryotes, and is crucial for fine-tuning the content of various metal elements required for plant metabolisms [12,14,15]. Notably, four Ferritin family genes (*AtFer1*-*AtFer4*) have been identified in *Arabidopsis*, and *AtFer1*, *AtFer3*, and *AtFer4* are highly expressed in leaves, whereas *AtFer2* is specifically expressed in seeds. Moreover, *AtFer1* is induced by excessive Fe and H_2_O_2_ stress, *AtFer2* is induced by abscisic acid (ABA) treatment, and *AtFer3* is induced by excessive Fe stress [14,15]. *Arabidopsis* Ferritin proteins are located in chloroplasts and form complexes with Fe^2+^, participating in the regulation of intracellular Fe^2+^ storage and sequestration, and maintaining plant tolerance to adverse environmental stresses, such as drought [16], water loss [17], and reactive oxygen species (ROS) [13,14,18]. The growth of the *Arabidopsis fer1fer3fer4* triple mutant was severely inhibited, and the intracellular Fe^2+^ content was sharply decreased. The *Arabidopsis fer2* mutant was very sensitive to ROS stress, and the seed germination rate was severely affected [13,14,18]. In addition, Ferritin is also involved in regulating the structure of roots. In the *Arabidopsis fer1fer3fer4* triple mutant, the destruction of ROS production and equilibration lead to changes in the structure of roots [18]. In recent years, ferritin homologous genes have been reported subsequently in plants, including soybean (*Glycine max*) [19], cut rose (*Rosa chinensis*) [17], and cassava (*Manihot esculenta*) [20]. However, biological functions of Ferritin proteins in fruit trees are essentially unknown.

Peach (*Prunus persica* L.) is a fruit that is popular worldwide, and its genome has been sequenced [21]. Among trace elements necessary for maintaining fruit tree growth and development, peach trees exhibit the highest demand for Fe, a factor closely tied to fruit quality and fruit yield [1,2,3]. In this study, a Ferritin family gene (*PpFer1*) was isolated from an elite peach variety, ‘Zhentong No. 3’, and differential responses of *PpFer1* to abiotic stresses (including Fe depletion, Fe toxicity, ABA, and H_2_O_2_ treatment) and the Fe^2+^ storage function were further determined. This study contributes to uncovering the molecular mechanisms of Fe storage and sequestration in fruit trees.

## 2. Materials and Methods

### 2.1. Plant Material and Growth Condition

The 7-year-old ‘Zhentong No. 3’ peach trees grown at the Zhenjiang Academy of Agricultural Sciences (Zhenjiang, China) were used in this study. Samples of leaves, stems and roots of seedlings, as well as bud-period flowers, full-blooming flowers, leaves, phloem, and fruit from both the young fruit stage (YFS) and mature fruit stage (MFS) of 7-year-old trees were collected and frozen in liquid nitrogen before qRT-PCR analysis. Biological replicates were conducted three times, each consisting of 15 distinct samples.

Tissue-cultured ‘Zhenjiang No. 3’ seedlings were germinated on half-strength MS solid medium (pH 5.8) for 1 month before being transferred to half-strength MS liquid solution in plastic incubators in a growth chamber [4,22]. For Fe depletion treatments, Fe was omitted from the half-strength MS liquid solution. For Fe toxicity treatments, seedlings were grown in half-strength MS liquid solution containing 500 μmol∙L^−1^ FeCl_3_ (pH 5.8). For ABA treatments, seedlings were grown in half-strength MS liquid medium supplied with 100 μmol∙L^−1^ ABA (pH 5.8), as previously described [17]. For oxidative stress treatments, seedlings were grown in half-strength liquid solution supplied with fresh H_2_O_2_ to a final concentration of 5% (*v*/*v*), as described in [17]. Seedlings were exposed to stress treatment for 48 h before expression analysis. Biological replicates were conducted three times, each involving 15 seedlings.

### 2.2. Physiological Analysis

The fresh weight of *Arabidopsis* seedlings was determined using a Thermo Electron Analytical Balance (Waltham, MA, USA). The roots of *Arabidopsis* seedlings were scanned using an Epson Rhizo scanner (Long Beach, CA, USA), and primary root length, lateral root numbers, and total root length were analyzed with the Epson WinRHIZO software 14.0 (Long Beach, CA, USA). *Arabidopsis* samples were digested using the HNO_3_-HClO_4_ method, and Fe concentration was assayed using ICP-AES systems (Thermo Electron, Waltham, MA, USA). The stomatal conductance (*G*_s_), net photosynthetic rate (*P*_n_), and transpiration rate (*T*_r_) were measured using a portable Li-COR Photosynthetic Apparatus (Lincoln, NE, USA) as previously described [4]. Chlorophyll was extracted using 95% ethanol and quantified using the BioRad SmartSpec 3000 spectrophotometer (Wadsworth, IL, USA), as previously mentioned [4]. 

### 2.3. Isolation and Cloning of PpFer Genes from Peach 

Taking the amino acid sequences of *Arabidopsis* AtFer1-4 as reference sequences [7,23], putative *PpFer* genes were obtained by screening the Peach Genome Database [21]. The genomic DNA sequence and coding sequence (CDS) of *PpFer* genes were downloaded (Table 1). The amino acid sequences of PpFer proteins were retrieved and verified regarding whether they possessed the Ferritin domain (PF00210) or not using the Pfam and InterProScan 4.8 online servers. Specific prime pairs were designed for CDS cloning of *PpFer* genes. The total RNA from 1-month-old ‘Zhentong No. 3’ seedlings was extracted using the RNAprep Pure Plant Kit (TianGen, Beijing, China) and synthesized into the first strand cDNA using the PrimeScript^TM^ RT reagent kit (Takara, Dalian, China). The CDSs of *PpFer* genes were amplified using the Prime STAR^TM^ HS DNA polymerase (Takara, Dalian, China) and further sequenced by Shenggong Bioengineering Co., Ltd. (Shanghai, China).

### 2.4. Phylogenetic Tree Construction

The alignment of amino acid sequences of Ferritin homologues from peach, *Vitis vinifera* (VvFer1-4), *A*. *thaliana* (AtFer1-4), *Arachis hypogaea* (AdFer1-4), *Camellia oleifera* (BnFer1-4), *Brassica rapa* (BrFer1-3), *Cicer arietinum* (CaFer1-3), *Gossypium hirsutum* (GhFer1-3), soybean (GmFer1-4), *Hevea brasiliensis* (HbFer2-4), *M*. *domestica* (MdFer3 and MdFer4), cassava (MeFer1-4), *Nicotiana tabacum* (NtFer1-2), *Ricinus communis* (RcFer2-3), *S*. *lycopersicum* (StFer1 and StFer2), and *Fragaria vesca* (FvFer3-4) was carried out using Cluster X 2.0.13 software. A phylogenetic tree of plant Ferritin homologues was constructed using the maximum likelihood method in MEGA 15.0, and a bootstrap test with 1000 replicates was performed to assess the confidence of the tree. 

### 2.5. Quantitative Real Time PCR (qRT-PCR) 

Specific primers for *PpFer* genes were designed using the NCBI/Primer-BLAST on-line server. Primer sequences are listed in Appendix A. PCR analysis was conducted on the 7500 Real Time PCR System (Applied Biosystems, New York, NY, USA), using the SYBR Premix Ex Taq (TaKaRa, Kyoto, Japan) reaction kit. The peach *Ubiquitin* gene served as the internal control, as established in previous studies [24,25]. Relative expression levels of *PpFer* genes were presented after normalization to the internal control *Ubiquitin*, based on three independent biological repeats, each with three technical replicates. 

To investigate the response of *PpFer* genes under abiotic stress treatments at the transcriptional level, the expression value under control conditions was set as 1. If the relative expression value under Fe depletion was <1, it indicated a decrease in gene expression level (depicted in blue). If the relative expression under Fe depletion value was >1, it signified an increase in gene expression level (depicted in red). The heat map of expression change was generated using the HemI software 18.3 [4,22].

### 2.6. Generation of Transgenic Arabidopsis Complementing PpFer1 Gene

The recombinant plasmid pBH-*PpFer1* was constructed by cloning the CDS of the *PpFer1* gene into the pBH vector [4,22]. This process utilized the forward primer of 5′-GACGGATCCATGCTTCTCAAAGGTTCTCC-3′ (*BamH* I underlined) and reverse primer of 5′-GAGTCTAGATCACGCAGCAATTGCATCAAC-3′ (*Xba* I underlined). The resulting recombinant plasmid was subcloned into *Agrobacterium tumefaciens* EHA 105 and subsequently transformed into the *Arabidopsis fer1-2* knockout homozygote mutant [26], which had been previously germinated on half-strength MS solid medium over 3 weeks, using the floral dip method. Independent T1 generations of *fer1-2*/35S::*PpFer1* complementation lines were obtained by screening hygromycin-resistant regenerated *Arabidopsis* seedlings. Genomic DNA was extracted from the T1 generation of *fer1-2/35S::PpFer1* lines using the Universal Genomic DNA Extraction Kit (TaKaRa, Dalian, China). The existence of an 846 bp product of *PpFer1* was further verified by reverse transcription PCR. T1 generation seedlings of *fer1-2/35S::PpFer1* were grown on half-strength MS solid medium for 2 weeks. Total RNA from shoots and roots of T1 transgenic lines was extracted using the RNAprep Pure Plant Kit (TianGen, Beijing, China) and synthesized into the first strand cDNA using the PrimeScript^TM^ RT reagent kit (Takara, Dalian, China) for the determination of *PpFer1* presence. Purified T3 generation seeds of #2 and #11 *fer1-2*/35S::*PpFer1* lines were harvested and sown on half-strength MS solid medium where they were kept for 7 days before physiological analysis. Biological replicates were conducted three times, each involving 30 seedlings.

### 2.7. Statistical Analysis

Graphs were generated using Origin 12.0 software, and significant differences were analyzed using Student’s *t*-test in SPSS 13.0 software (SPSS Chicago, IL, USA) or Fisher’s LSD test in the ANOVA software 13.0, with details provided in the legends.

## 3. Results

### 3.1. Isolation of Ferritin Genes in Peach

In total, three putative *Ferritin* genes were identified from the peach genome, which were named *PpFer1*-*PpFer3* (Table 1 and Figure 1). Verification of the protein domain demonstrated that all of the PpFer proteins exhibit the Ferritin domain (PF00210), indicating that all of them are Ferritin transporters (Figure 1). The percentage of amino acid sequence identities among peach Ferritins was 70.82% (Figure 1). The percentage of amino acid sequence identities among peach Ferritins and homologues from 15 other plants was 56.65% (Appendix A). 

Phylogenetic tree analysis showed that PpFer1, PpFer2, and PpFer3 were tightly clustered with strawberry FvFer4, grape VvFer4, and apple MdFer4, respectively, implying that PpFer transporters possess relatively close evolutionary distance from *Rosaceae* homologues (Figure 2). 

### 3.2. Expression Profiles of PpFer Genes

Results showed that the expression levels of *PpFer* genes were quite distinct among different tested tissues, including different tissues of ‘Zhentong No. 3’ seedlings, flowers in both bud period and full blooming stage, and annual leaves, phloem, and fruits from both the young fruit stage and mature fruit stage (Figure 3). Notably, the overall expression of *PpFer1* was the most abundant, followed by *PpFer3*, with *PpFer2* being specifically expressed in fruit from the young fruit stage (YFS). In addition, the highest expression level of *PpFer1* was observed in leaves from the mature fruit stage (MFS), followed by phloem from the YFS and full-bloom flowers, and the highest expression level of *PpFer3* was found in roots of seedlings, followed by phloem, fruit and leaves from the YFS (Figure 3).

### 3.3. Differential Response of PpFer Genes under Abiotic Stress Treatment in Tissue-Cultured Seedlings

Further analysis showed that *PpFer* genes responded differentially to abiotic stresses, including Fe depletion, Fe toxicity, ABA stress, and H_2_O_2_ stress, in tissue-cultured peach seedlings (Figure 4). In detail, *PpFer1* was quite sensitive to Fe toxicity and H_2_O_2_ treatment, and its expression levels were up-regulated throughout the whole plant seedlings. *PpFer3* responded to Fe toxicity and ABA treatment, and its expression levels were significantly increased in all tested tissues (leaves, stems, or roots). However, expression of *PpFer2* changed little in all tested tissues under every treatment in this study (Figure 4).

### 3.4. PpFer1 Rescued the Retarded Growth of Arabidopsis fer1-2 Mutant

In *Arabidopsis*, growth of *fer1-2* knockout mutant was hindered, accompanied by chlorosis symptoms [26]. To determine whether grape *PpFer1* could restore the normal growth of *fer1-2* mutant, *PpFer1* was subcloned into the binary expression vector pHB (Appendix A). At least seven putative (#1, #10, #11, #12, #14, #15, and #16) T1 generation *fer1-2*/35S::*PpFer1* complementation lines were verified using reverse transcription PCR for the presence of a 846 kb fragment of *PpFer1* (Appendix A). Purified T3 generation of #1 and #10 *fer1-2*/35S::*PpFer1* lines were randomly selected for further physiological analysis. 

Compared with control conditions, growth of both wild type and *fer1-2* mutant lines was decreased under Fe depletion, Fe toxicity, ABA treatment, or H_2_O_2_ treatment, which was embodied in reduced total fresh weight, primary root length, and lateral root numbers (Figure 5 and Figure 6). Compared with the wild type, growth of *fer1-2* mutant lines was hindered under control conditions, Fe toxicity, ABA treatment, or H_2_O_2_ treatment (Figure 5), accompanied by decreased fresh weight (Figure 6A), primary root length (Figure 6B), and lateral root numbers (Figure 6C). However, no growth difference was observed between *fer1-2* mutant and the wild type under Fe depletion treatment (Figure 6). 

The #1 and #10 lines were paired with different wild type seedlings. We conducted statistical analysis on both the #1 and #10 lines, and an identical trend was observed. Data from the #1 *fer1-2*/35S::*PpFer1* lines are presented in this study (Figure 6). Notably, growth of #1 *fer1-2*/35S::*PpFer1* lines were significantly strengthened compared with that of *Arabidopsis fer1-2* mutant lines under control conditions, Fe toxicity, or H_2_O_2_ treatment, as evidenced by increased total fresh weight, primary root length, and lateral root numbers, which were similar to those of the wild type (Figure 5 and Figure 6). These findings indicate that over-expression of *PpFer1* successfully restored the retarded growth of *Arabidopsis fer1-2* mutant lines under control conditions, Fe toxicity, or H_2_O_2_ treatment. However, the growth of the #1 *fer1-2*/35S::*PpFer1* lines remained the same as that of the *fer1-2* mutant lines under ABA treatment or Fe depletion conditions. 

In addition, the tissue Fe concentration (Figure 6D), total leaf chlorophyll (Figure 6E), *P*_n_ (Figure 6F), *G*_s_ (Figure 6G), and *T*_r_ (Figure 6H) of *fer1-2* mutant lines was reduced under control conditions, Fe toxicity, H_2_O_2_ treatment, or ABA treatment but changed little under Fe depletion, compared with that of wild type lines (Figure 6). In particular, both the #1 and #10 *fer1-2*/35S::*PpFer1* lines exhibited higher tissue Fe concentration, total leaf chlorophyll, *P*_n_, *G*_s_, and *T*_r_ than *fer1-2* mutant lines under control conditions, Fe toxicity, and H_2_O_2_ treatment, but changed little under Fe depletion or ABA treatment (Figure 6).

## 4. Discussion

In fruit trees, Fe is one of the most indispensable mineral elements. It directly affects tree growth, flowering, fruit quality formation, and fruit yield [1,3,4,24,25]. Currently, the effective Fe concentration in natural soils does not correspond with normal growth of fruit trees under normal pH values [1,3,7]. However, molecular mechanisms towards Fe uptake, transport, distribution, and storage in fruit trees are essentially unknown. In particular, peach is a dicotyledonous fruit tree that belongs to the Mechanism I Fe absorption category of plants [7,10,11]. In this study, three Ferritin transporters were isolated from peach. These transporters are prone to being closely clustered with *Rosaceae* homologues, implying that peach Ferritins may possess a close genetic distance and similar biological functions to *Rosaceae* fruit trees, as a result of long-term evolution. Therefore, studying the biological function of peach Ferritin transporters contributes to revealing the biological function of Ferritin homologues from *Rosaceae* fruit trees.

In this study, *PpFer1* and *PpFer3* could be detected in all tested tissues but *PpFer3* is specifically expressed in young-stage fruit. Notably, *PpFer1* is highly expressed in leaves, which was in line with *AtFer1*, *AtFer3*, and *AtFer4* in *Arabidopsis*, *PbFer2* in pear [27], and *MeFer4* in cassava [20]. *PpFer1* is also highly expressed in full-bloom flowers, which is similar to *RhFer* in cut rose [17], whereas *PpFer2* is exclusively expressed in young peach fruit, *Arabidopsis AtFer2* is specifically expressed in roots, and tomato *SlFer* is majorly expressed in root tips. These findings suggest that Ferritin transporters possess extensive expression profiles and some of them are likely to be functional in specific tissues or organs in plants. 

Previous studies have demonstrated that *AtFer1* and *AtFer3* are induced by excessive Fe and H_2_O_2_ stress, and AtFer2 is not responsive to iron [23]. Consistently, *PpFer1* and *PpFer3* were responsive to excessive Fe and H_2_O_2_, and their expression was significantly up-regulated throughout the whole seedling. These findings imply that *PpFer1* and *PpFer3* are likely to be active in regulating the Fe storage capacity in peach cells under Fe toxicity conditions or reactive oxygen species stress, thus maximally maintaining the cytosol Fe concentration in moderation so as to secure the basic life activities depending on Fe. Simultaneously, the Fe concentration was enhanced under Fe toxicity but reduced under H_2_O_2_ treatment in all tested *Arabidopsis* lines, which contributes to securing the basic growth of peach seedlings. In pear, *PbFer2* is inhibited by iron deficiency stress [27]. However, *PpFer* genes did not respond to Fe depletion in this study. We speculate that *PpFer* genes are prone to being active in storing Fe in peach trees under excessive Fe conditions but not Fe deficiency conditions. *PpFer2* was very much less expressed in tested peach tissues/organs and had little response to any abiotic treatment in this study. We hypothesize that this gene is likely to be a pseudogene that may have lost its protein-coding ability due to accumulated mutations or unprocessed segmental duplication over long-term evolution [28]; this idea requires further verification. 

Expression of *AtFer2* in *Arabidopsis* [16,23] and *RhFer* in cut rose [17] was induced by ABA treatment, which was also observed in *PpFer3* in this study. Indeed, ABA treatment induced the cellular Fe accumulation in all tested *Arabidopsis* lines, which was in accord with previous studies in transgenic tomato [29]. Nonetheless, these genes regulated by both Fe and ABA may play a role in the crosstalk between Fe and ABA, and may be an intermediate hub for the cross-linking of iron and ABA signals. 

In *Arabidopsis*, *AtFer1* regulates the free Fe levels in plant cells and knockout of *AtFer1* accelerated natural senescence of *Arabidopsis* seedlings with hindered growth status [26]. As the most abundantly expressed *Ferritin* gene in grape, the maximum expression of *PpFer1* was detected in aboveground parts of peach trees and was increased in all tested tissues under excessive Fe and H_2_O_2_ stress. Favorably, the over-expression of *PpFer1* was effective in restoring the retarded growth of *Arabidopsis fer1* knockout mutant. Notably, tissue Fe concentration and photosynthesis performance were significantly strengthened in *fer1-2*/35S::*PpFer1* lines, which may partially explain the rescued growth status. The complementation of *PpFer1* may actively mobilize the Fe storage capacity of *fer1-2*/35S::*PpFer1* lines, thereby preventing Fe toxicity or oxidative stress, which helps maintain basic life activities or metabolic processes that rely on a moderate Fe level. Synchronously, total leaf chlorophyll, *P*_n_, *G*_s_, and *T*_r_ were increased in *fer1-2*/35S::*PpFer1* lines. Considering that *MeFer4* is induced by low temperature, and transgenic lines with *MeFer4* over-expression have enhanced cold resistance in cassava [20], we speculate that Ferritin transporters are implicated in enhancing plant resistance to undesired abiotic stresses, including cold, Fe toxicity, and H_2_O_2_ stress. These findings again support the proposition that Ferritin transporters control interaction between Fe homeostasis and oxidative stress in plants [13,17,20]. Nonetheless, PpFer1 may be an important Ferritin transporter that is involved in Fe storage in peach, especially under excessive Fe or oxidative stress conditions.

## 5. Conclusions

Three *PpFer* genes were isolated from peach; their expression levels were significantly different in distinct tissues/organs. *PpFer1* was the most abundantly expressed *Ferritin* gene in peach; its expression was induced under excessive Fe and H_2_O_2_ stress in all tested tissues. Over-expression of *PpFer1* rescued the retarded growth of *Arabidopsis fir1-2* knockout mutant, which was embodied in strengthened fresh weight, primary root length, total root length, total leaf chlorophyll, *P*_n_, *G*_s_, *T*_r_, and tissue Fe concentration.

## Figures and Tables

**Figure 1 plants-12-04093-f001:**
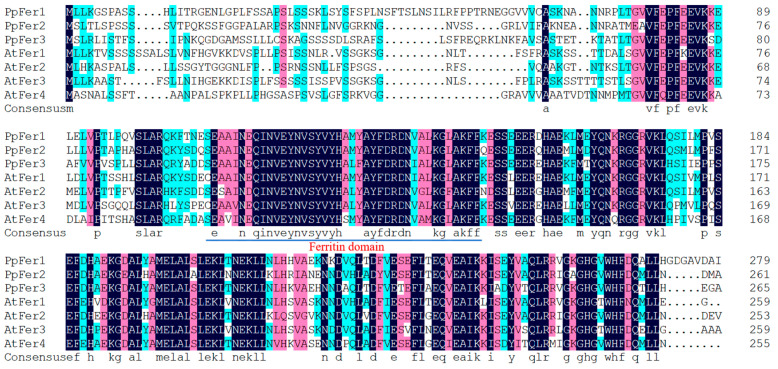
Amino acid alignment of Ferritin proteins from peach and *Arabidopsis*. The color of black, pink, and dark green indicates the identity of 100%, 85%, and the range between 45% and 70%, respectively, at the same amino acid residue.

**Figure 2 plants-12-04093-f002:**
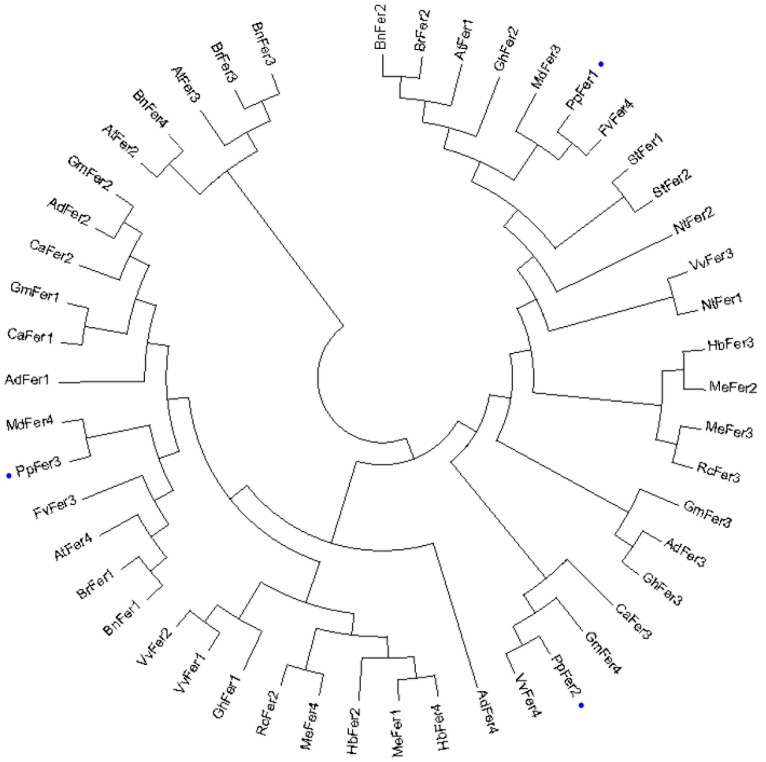
Phylogenetic tree of plant Ferritin homologues. The alignment of amino acid sequences of Ferritin homologues from peach (PpFer1-3), *Vitis vinifera* (VvFer1-4), *Arabidopsis thaliana* (AtFer1-4), *Arachis hypogaea* (AdFer1-4), *Camellia oleifera* (BnFer1-4), *Brassica rapa* (BrFer1-3), *Cicer arietinum* (CaFer1-3), *Gossypium hirsutum* (GhFer1-3), *Glycine max* (GmFer1-4), *Hevea brasiliensis* (HbFer2-4), *M*. *domestica* (MdFer3 and MdFer4), *Manihot esculenta* (MeFer1-4), *Nicotiana tabacum* (NtFer1-2), *Ricinus communis* (RcFer2-3), *Solanum lycopersicum* (StFer1 and StFer2), and *Fragaria vesca* (FvFer3-4) was carried out using Cluster X 2.0.13 software. A phylogenetic tree was constructed using the maximum likelihood method in MEGA 15.0. Grape PpFer proteins are labelled with blue dots.

**Figure 3 plants-12-04093-f003:**
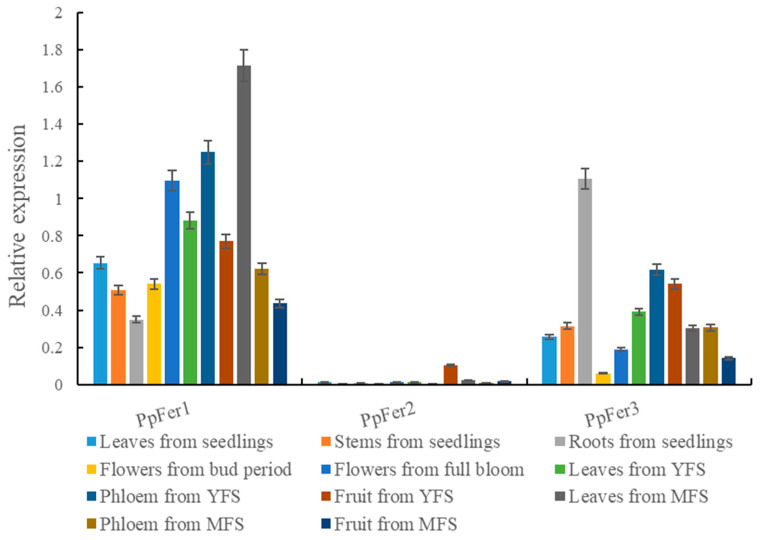
Tissue-specific expression analysis of *PpFer* genes. Tissue samples from tissue-cultured seedlings, and young leaves, mature leaves, full blooming flowers, young fruits, and mature fruits from 7-year-old ‘Zhentong No. 3’ trees were collected on specific dates of 2021, and frozen immediately in liquid nitrogen before qRT-PCR analysis. YFS, young fruit stage. MFS, mature fruit stage.

**Figure 4 plants-12-04093-f004:**
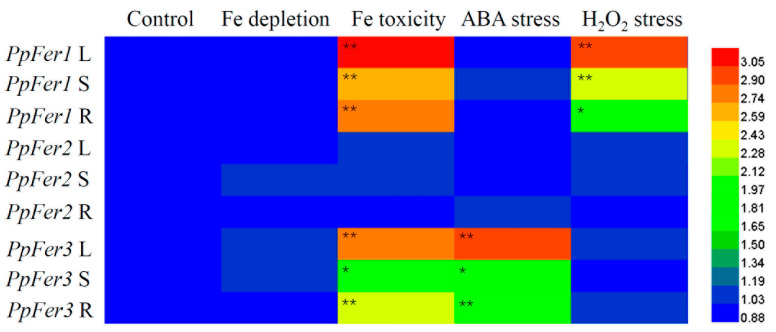
Response of *PpFer* genes to abiotic stress treatments in tissue-cultured seedlings. One-month-old tissue-cultured ‘Zhentong No. 3’ seedlings were subjected to iron depletion, Fe toxicity (500 μmol∙L^−1^ FeCl_3_, pH 5.8), 100 μmol∙L^−1^ ABA (pH 5.8), or 5% (*v*/*v*) H_2_O_2_ (pH 5.8) treatment for 48 h before qRT-PCR analysis. Relative expression levels of *PpFer* genes were presented after normalization to *Ubiquitin* (the internal control) from three independent biological replicates. Asterisks indicate statistical differences found between the control and abiotic stress treatment using Student’s *t*-test in SPSS 13.0 software (* *p* < 0.05, ** *p* < 0.01).

**Figure 5 plants-12-04093-f005:**
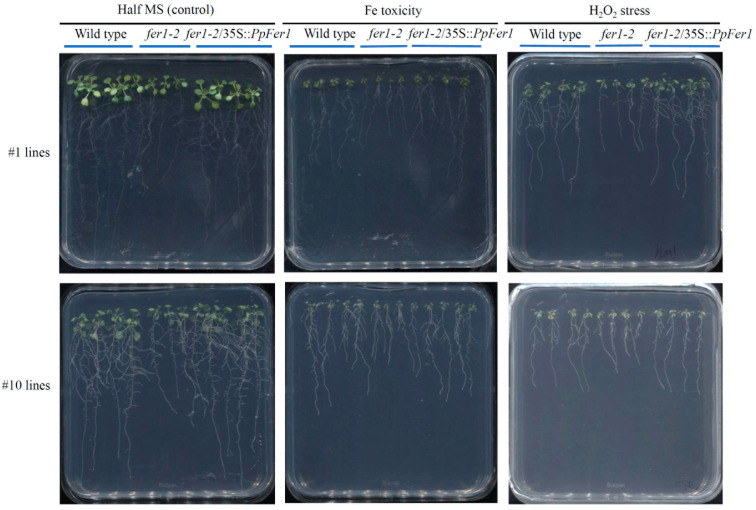
Phenotype analysis of *PpFer1* over-expression transgenic *Arabidopsis* seedlings. T3 generation seeds of #1 and #10 lines were germinated on half-strength MS solid medium, and then subjected to 100 μmol∙L^−1^ ABA (pH 5.8) or 5% (*v*/*v*) H_2_O_2_ (pH 5.8) treatment for 7 days before phenotype analysis.

**Figure 6 plants-12-04093-f006:**
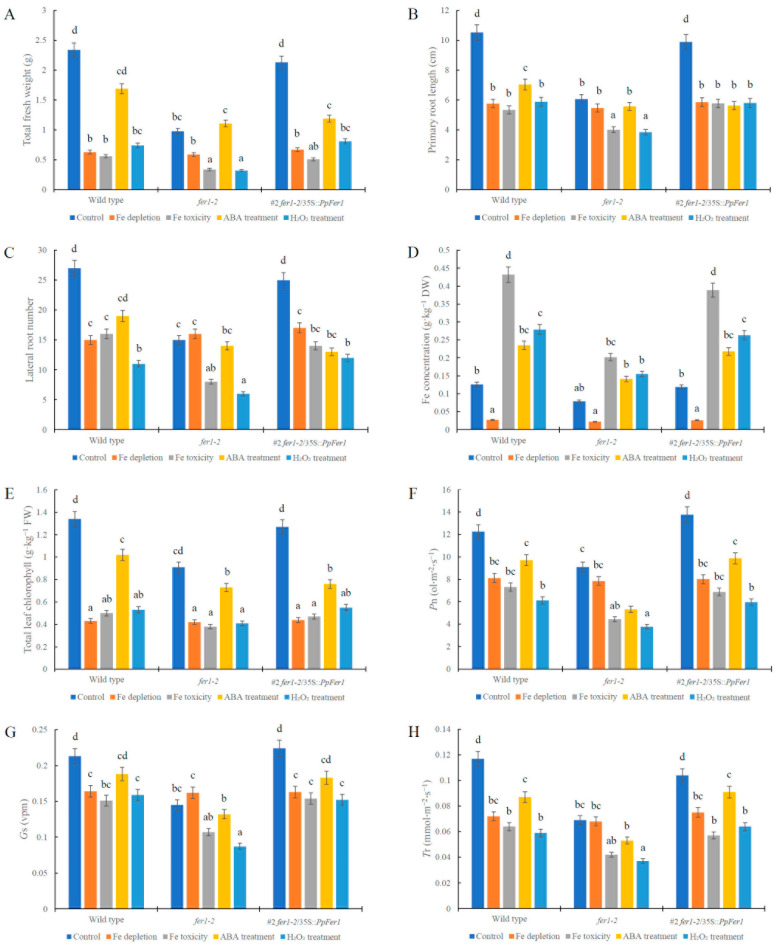
Physiological analysis of *PpFer1* over-expression in transgenic *Arabidopsis* seedlings. (**A**) Total fresh weight. (**B**) Primary root length. (**C**) Lateral root number. (**D**) Fe concentration. (**E**) Total leaf chlorophyll. (**F**) *P*_n_. (**G**) *G*_s_. (**H**) *T*_r_. T3 generation seeds of #1 lines were germinated on half-strength MS solid medium and then subjected to 100 μmol∙L^−1^ ABA (pH 5.8) or 5% (*v*/*v*) H_2_O_2_ (pH 5.8) treatment for 7 days before physiological analysis. Data are presented as means ± SE (*n* = 30). Letters indicate statistical differences between the wild type, *fer1-2*, and *fer1-2*/35S::*PpFer1* lines under all treatments using Fisher’s LSD test in ANOVA software 13.0.

**Table 1 plants-12-04093-t001:** *PpFer* gene information.

Gene	Locus No.	Location	Chromosome Distribution	CDS (bp)
*PpFer1*	Prupe.2G256100	Pp02:26807278..26810901 reverse	Chr02	846
*PpFer2*	Prupe.6G050900	Pp06:3570278..3572650 forward	Chr06	786
*PpFer3*	Prupe.6G283700	Pp06:26383839..26386074 reverse	Chr06	798

## Data Availability

The data are contained within the present article and in its Appendix A.

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
