# Peer review of "Heterologous Expression of a Ferritin Homologue Gene PpFer1 from Prunus persica Enhances Plant Tolerance to Iron Toxicity and H2O2 Stress in Arabidopsis thaliana"

_plants, 2023, doi:10.3390/plants12244093_

Round 1

Reviewer 1 Report

Comments and Suggestions for Authors

The authors in this study present data on the biological roles of ferritin genes of Prunus persica trees using molecular, physiological and genetic approaches. They identify three candidate ferritin genes in Prunus based on their homology with Arabidopsis protein sequences. They study the tissue-specific expression of those gene in prunus trees, showing that PpFer1 is the most abundant. Then they induce various stresses in in vitro grown prunus seedlings to show that PpFer1 and PpFer3 respond to some of those stresses. Finally they use ferritin mutants of Arabidopsis and compliment those with PpFer1 gene showing that this gene functions with the same way in Arabidopsis thaliana.

Some commends/issues with the manuscript:

line 19-20 difficult to understand, please rephrase

line 75 species name should be in lower case letter

line 83 what is this line for?

line 179 the authors should describe better how the genes where identified

line 184  "...data not shown..." all data should be presented at least as supplementary material

line 190 "..prone to possess..." that is a weird way to phrase this

the authors should explain more what is the meaning of the phylogenetic relationships in Fig.2

line 207 what is YSF? please explain what that stands for

Fig1. This figure will benefit if the Arabidopsis sequences are also shown; also some information about the domains of the protein, if available, could be shown

Fig2. this is ok, maybe better resolution

Fig3. ok, what MFS, YFS stands for?

Fig.4 the methods used to collect the data for this figure should be shown, also there are no asterisks showing the statistically significant values

Fig5. this whole figure could go to the supplementary data

I would also like to see the transgenic plants in soil at a later stage e.g rosette or even with mature siliques to show the phenotype of the transgenic plants throughout development

The expression of the trangenes could be assessed by q-RT-PCR in order to identify differences in the level of expression of each line that could affect the phenotype

Fig.6 Better quality photos will benefit this figure

Fig7. In this figure the authors assess only one trangenic line, the line #2.

Since this is the main point of the whole study, i believe at least a second line should be analysed. Also, there is no explanation about the letters that show the statistically significant values.

Finally the manuscript would greatly benefit if the authors try to complement the arabidopsis fer mutants with the other 2 prunus genes identified.

Comments on the Quality of English Language

Some proof-reading in the language of the manuscript should be performed, since some sentences are hard to understand.

Reviewer 2 Report

Comments and Suggestions for Authors

Overall this paper is well written and has a thorough method that complements the results of the paper. All experiments are well designed and the data supports the conclusions of the paper. The statistics and analysis of the data are sound. The materials and methods are appropriate and relevant to the results of the data. The discussion and conclusions are impactful to the field. This paper identifies three homologs of Ferritin in peach trees. The paper's notable outcomes of the research are: the location of expression of the Ferritin homologs in the various tissues of the peach tree, the proof of function by complementing Arabidopsis ferritin mutants and the induction of the genes by both iron and oxidative stress. 

Comments on the Quality of English Language

There are a few places where the plural of the word should be used and not the singular version. But overall only minor editing of English is needed. The chemical formulas should be checked to make sure that they have the numbers listed as subscripts. Also with some H2O2 text in the paper the second O2 is a zero (0) and not an "O", i.e. H202 and not the proper H2O2. This needs to be fixed. Examples can be found in lines 220, 228, 246, 254, 260,  266, 272, 275 The degrees Celsius symbol oC throughout the paper looks like a different font than the rest of the text.

Round 2

Reviewer 1 Report

Comments and Suggestions for Authors

In the revised form of the manuscript, the authors covers most of the reviewers' comments. The manuscript has been significantly improved and the mistakes were rectified.

Comments on the Quality of English Language

 i believe there are a couple of spelling mistakes here and there, some further spell-proofing could be needed.